# Multi-tissue transcriptome-wide association study identifies eight candidate genes and tissue-specific gene expression underlying endometrial cancer susceptibility

Pik Fang Kho [1,2], Xuemin Wang[1], Gabriel Cuéllar-Partida [1,3], Thilo Dörk [4], Ellen L. Goode[5], Diether Lambrechts[6], Rodney J. Scott [7,8], Amanda B. Spurdle [1], Tracy A. O'Mara [1,9✉] & Dylan M. Glubb [1,9]

Genome-wide association studies (GWAS) have revealed sixteen risk loci for endoemtrial cancer but the identification of candidate susceptibility genes remains challenging. Here, we perform transcriptome-wide association study (TWAS) analyses using the largest endometrial cancer GWAS and gene expression from six relevant tissues, prioritizing eight candidate endometrial cancer susceptibility genes, one of which (*EEFSEC*) is located at a potentially novel endometrial cancer risk locus. We also show evidence of biologically relevant tissue-specific expression associations for *CYP19A1* (adipose), *HEY2* (ovary) and *SKAP1* (whole blood). A phenome-wide association study demonstrates associations of candidate susceptibility genes with anthropometric, cardiovascular, diabetes, bone health and sex hormone traits that are related to endometrial cancer risk factors. Lastly, analysis of TWAS data highlights candidate compounds for endometrial cancer repurposing. In summary, this study reveals endometrial cancer susceptibility genes, including those with evidence of tissue specificity, providing insights into endometrial cancer aetiology and avenues for therapeutic development.

[1] Department of Genetics and Computational Biology, QIMR Berghofer Medical Research Institute, Brisbane, QLD, Australia. [2] School of Biomedical Sciences, Queensland University of Technology, Brisbane, QLD, Australia. [3] 23andMe Inc, Sunnyvale, CA, USA. [4] Gynaecology Research Unit, Hannover Medical School, Hannover, Germany. [5] Department of Health Science Research, Division of Epidemiology, Mayo Clinic, Rochester, MN, USA. [6] Laboratory for Translational Genetics, Department of Human Genetics, KU Leuven, VIB Center for Cancer Biology, Leuven, Belgium. [7] Division of Molecular Medicine, Pathology North, John Hunter Hospital, Newcastle, NSW, Australia. [8] Discipline of Medical Genetics, School of Biomedical Sciences and Pharmacy, Faculty of Health, University of Newcastle, Newcastle, NSW, Australia. [9] These authors contributed equally: Tracy A. O'Mara, Dylan M. Glubb. ✉email: Tracy.OMara@qimrberghofer.edu.au

Endometrial cancer is the most commonly diagnosed gynaecological cancer in developed countries, with incidence expected to continue to increase over the next decade (reviewed in Morice et al.[1]). To explore the role of common genetic variation in endometrial cancer susceptibility, we recently conducted a genome-wide association study (GWAS) meta-analysis and identified 16 susceptibility loci[2]. Despite the success of GWAS in identifying genetic variants underlying endometrial cancer risk, biological interpretation remains challenging given that the majority of the GWAS risk variation is located in non-protein-coding regions (reviewed in O'Mara et al.[3]).

Trait-associated loci identified by GWAS have demonstrated enrichment for expression quantitative trait loci (eQTLs), suggesting that changes in gene expression link genetic variation and phenotypes[4]. As eQTL data are not typically available for GWAS samples, the transcriptome-wide association study (TWAS) approach has been developed to integrate eQTL and GWAS data from independent sample sets[5]. TWAS uses effect estimates from eQTLs to impute gene expression in GWAS samples, and subsequently associate the imputed gene expression with phenotypes. TWAS has an additional advantage of alleviating the multiple testing penalty of GWAS in statistical inference by testing the imputed expression of thousands of genes rather than millions of genetic variants.

TWAS has identified candidate causal genes for many complex traits, using eQTL data from disease-relevant tissues or cell types[6–9]. However, endometrial or uterine eQTL data are less commonly available than for other tissues or cell types (e.g., whole blood). Nevertheless, Mortlock et al.[10] have shown that 85% of eQTLs found in the endometrium are shared across multiple tissues, suggesting that data from other tissues could be incorporated to perform endometrial cancer TWAS. Recently, innovative multi-tissue TWAS approaches that leverage shared eQTLs across multiple tissues have become available, and provide increased statistical power to identify candidate causal genes[11,12]. These multi-tissue TWAS approaches account for the correlation of gene expression between tissues and perform multivariate regression to predict gene expression from multiple tissues simultaneously.

In the current study, we identify eight candidate endometrial cancer susceptibility genes by conducting multi-tissue TWAS analyses, integrating GWAS data from the Endometrial Cancer Association Consortium[2] and eQTLs from six related or biologically relevant tissues from the Genotype-Tissue Expression (GTEx) project[4]: two adipose tissue types (subcutaneous adipose and visceral omentum adipose), three gynaecological tissues (ovary, uterus and vagina) and whole blood. Two of the candidate susceptibility genes (AC021755.3 and EEFSEC) have not been previously reported and EEFSEC is located outside established endometrial cancer GWAS risk loci. Notably, follow-up analyses suggest evidence of tissue-specific gene expression associations. We also perform a phenome-wide association analysis to uncover traits that relate to candidate endometrial cancer susceptibility genes and provide insights into endometrial cancer aetiology. Finally, we explore drug repurposing opportunities for endometrial cancer by analysing TWAS findings using the Connectivity Map database[13] and Open Target platform[14], and highlight candidate compounds for endometrial cancer treatment

expression associated with endometrial cancer susceptibility (Table 1). Given that the TWAS associations could result from linkage disequilibrium (LD) between genetic variants independently affecting gene expression and endometrial cancer risk, we performed colocalization analysis to reduce LD contamination[15]. The associations of CYP19A1, SKAP1, HEY2, EEFSEC, and EVI2A were supported by colocalization of eQTL and GWAS signals in at least one of the relevant tissues (Table 1); thus, these five genes were prioritized as candidate endometrial cancer susceptibility genes. The colocalization data also suggested tissue-specific associations for CYP19A1 (subcutaneous adipose), HEY2 (ovary) and SKAP1 (whole blood) (Table 1).

The second multi-tissue TWAS analysis using joint-tissue imputation (JTI)[16] revealed 11 unique genes whose imputed expression associated with endometrial cancer susceptibility in at least one of the six tissues studied (Table 2). Five of the seven genes (CYP19A1, SKAP1, HEY2, SNX11, and EEFSEC) initially revealed by S-MultiXcan (i.e. before prioritisation by colocalization analysis) were also identified using JTI. To prioritise the genes identified by JTI, we performed a Mendelian randomization (MR) causal inference analysis in the respective tissues using MR-JTI, providing five candidate endometrial cancer susceptibility genes: CYP19A1, SNX11, AC021755.3, EEFSEC, and EIF2AK4 (Table 2). These analyses indicated some tissue specificity in the predicted gene expression associations: CYP19A1 in adipose (subcutaneous and visceral omentum), AC021755.3 in uterus, and EEFSEC and EIF2AK4 in vagina. However, the colocalization analysis of the EEFSEC S-MultiXcan result suggested the involvement of other tissues (Table 1). Two candidate endometrial cancer susceptibility genes (CYP19A1 and EEFSEC) were prioritized using both S-MultiXcan-colocalization and MR-JTI approaches.

Secondary TWAS analysis of the endometrioid endometrial cancer subtype using S-MultiXcan identified five genes, three of which (CYP19A1, HEY2, and SKAP1) passed the colocalization threshold (Supplementary Table 1); all three genes had also been identified in the main S-MultiXcan-colocalization analysis using all endometrial cancer cases (Table 1). Analysis of endometrioid endometrial cancer using JTI identified seven unique genes (Supplementary Table 2). Subsequent causal inference test by MR-JTI revealed two candidate genes for endometrioid endometrial cancer susceptibility (SNX11 and EIF2AK4 (Supplementary Table 2)), both of which had been identified in the main MR-JTI analysis using all endometrial cancer cases (Table 2). Thus, the two multi-tissue TWAS and associated analyses prioritized eight candidate endometrial cancer susceptibility genes: AC021755.3, EEFSEC, EIF2AK4, CYP19A1, HEY2, SNX11, EVI2A, and SKAP1. Gene expression of the candidate susceptibility genes was generally positively correlated across the solid tissues studied (Fig. 1a–h). However, gene expression in whole blood was occasionally inversely, or less well correlated, with expression levels in other tissues (e.g., EIF2AK4 (Fig. 1c), EVI2A (Fig. 1d), and SNX11 (Fig. 1h)).

**Phenome-wide association study (PheWAS) analysis of candidate susceptibility genes highlights associations with traits related to endometrial cancer risk.** PheWAS analysis of the eight candidate susceptibility genes revealed that four candidate genes (CYP19A1, EIF2AK4, EVI2A and SNX11) associated with 56 traits related to seven phenotypic categories: anthropometric, bone health, cardiovascular, diabetes, hematopoiesis, liver function, and sex hormones (Supplementary Table 3). Five of these categories (anthropometric, bone health, cardiovascular, diabetes, and sex hormones) contain traits that relate to or include genetically established endometrial cancer risk factors (e.g. body mass

## Results
**Multi-tissue TWAS analyses identify eight candidate susceptibility genes for endometrial cancer.** We performed two multi-tissue TWAS analyses using eQTL data from six related or biologically relevant tissues. The first multi-tissue TWAS analysis using S-MultiXcan[11] revealed seven genes whose imputed

**Table 1 Genes significantly associated with endometrial cancer risk as identified by S-MultiXcan and colocalization.**

| Locus | Gene | $N_{tissue}$ | P-value | $Z_{min}$ | $Z_{max}$ | $Z_{mean}$ | $Z_{SD}$ | $PP_{Adipose Sub.}$ | $PP_{Adipose Vis.}$ | $PP_{Ovary}$ | $PP_{Vagina}$ | $PP_{Uterus}$ | $PP_{WB}$ |
|---|---|---|---|---|---|---|---|---|---|---|---|---|---|
| 3q21.3 | EEFSEC | 3 | 1.10E−06 | 2.98 | 5.07 | 4.08 | 1.05 | **0.85** | **0.85** | 0.01 | 0.09 | 0.02 | **0.88** |
| 6q22.31 | HEY2 | 1 | 9.94E−09 | −5.73 | −5.73 | −5.73 | NA | 0.00 | 0.00 | **0.89** | 0.01 | 0.02 | 0.00 |
| 15q21.2 | GLDN | 4 | 1.34E−12 | −2.02 | 3.65 | −0.23 | 2.62 | 0.00 | 0.00 | 0.01 | 0.02 | 0.00 | 0.00 |
| 15q21.2 | CYP19A1 | 2 | 9.52E−12 | 5.75 | 6.17 | 5.96 | 0.30 | **0.82** | 0.02 | 0.00 | 0.01 | NA | 0.28 |
| 17q11.2 | EVI2A | 3 | 1.50E−06 | −5.14 | −3.45 | −4.51 | 0.92 | **0.90** | **0.88** | 0.03 | 0.02 | 0.01 | **0.88** |
| 17q21.32 | SKAP1 | 2 | 7.27E−09 | −6.11 | −0.15 | −3.13 | 4.22 | 0.01 | 0.01 | 0.01 | 0.01 | 0.01 | **0.95** |
| 17q21.32 | SNX11 | 4 | 5.41E−07 | 3.49 | 5.38 | 4.63 | 0.83 | 0.03 | 0.03 | 0.03 | 0.02 | 0.03 | 0.02 |

The Bonferroni corrected P value threshold is $3.8 \times 10^{-6}$ (accounting for 13,182 genes tested). Bolded findings have evidence of colocalization (PP > 0.8) in at least one tissue. $N_{tissue}$ number of tissues with available expression weights, $Z_{min}$ minimum Z score from single tissue S-PrediXcan result, $Z_{max}$ maximum Z score from single tissue S-PrediXcan result, $Z_{mean}$ mean Z score from single tissue S-PrediXcan result, $Z_{SD}$ standard deviation of Z score from single tissue S-PrediXcan result, PP posterior probability that endometrial cancer risk and eQTL variants from respective tissue colocalize, Adipose Sub. subcutaneous adipose, Adipose Vis. visceral omentum adipose, WB whole blood, NA not available.

index[2], LDL and HDL cholesterol[17], insulinemia[18], estradiol[19], sex hormone-binding globulin (SHBG) and testosterone[20]). Of the PheWAS traits with available GWAS data, 13 demonstrated evidence of a significant genetic correlation with endometrial cancer, including SHBG, testosterone, the impedance of arm (related to body mass index) and apolipoprotein A (a key constituent of HDL) (Supplementary Table 4; Supplementary Fig. 1).

**Drug repurposing analysis reveals candidates for endometrial cancer therapy**. We integrated data from S-MultiXcan and JTI analyses with drug-induced gene expression profiles from the Connectivity Map database, identifying four compounds with profiles opposing the expression of genes associated with endometrial cancer risk (Table 3). None of these drugs have been approved for endometrial cancer treatment; however, enzastaurin has anti-cancer effects across endometrial cancer cell lines (mean $IC_{50} = 30\,\mu M$) in the Genomics of Drug Sensitivity in Cancer database[21]. We also explored drug repurposing opportunities using the Open Targets platform to assess the eight candidate susceptibility genes. Only the protein encoded by CYP19A1 (aromatase) is targeted by clinically approved drugs, with indications for breast cancer and Cushing syndrome (Supplementary Table 5).

## Discussion

Two multi-tissue TWAS approaches prioritized eight candidate susceptibility genes for endometrial cancer. Two of these genes (CYP19A1 and EEFSEC) were identified by both sets of analyses and four candidate susceptibility genes (AC021755.3, CYP19A1, EIF2AK4, HEY2, and SKAP1) demonstrated evidence of tissue specificity. Secondary analysis revealed that five of the eight candidate genes (CYP19A1, EIF2AK4, HEY2, SKAP1, and SNX11) were also candidate susceptibility genes for the endometrioid endometrial cancer subtype. All candidate susceptibility genes except for EEFSEC (3q21.3) are located at known endometrial cancer GWAS risk loci. EEFSEC encodes a eukaryotic elongation factor that is involved in the incorporation of selenocysteine into proteins. This gene has been shown to be necessary for the viability for a number of cancer cell lines, including two derived from endometrial tumors (https://depmap.org/portal/), but its specific function in endometrial carcinogenesis is unknown. The 3q21.3 locus has been preliminarily reported as a risk locus for endometrial cancer in a cross-cancer GWAS meta-analysis[22] and may provide a genome-wide significant risk loci in future larger endometrial cancer GWAS.

Three of the TWAS candidate susceptibility genes (EIF2AK4, SKAP1, and SNX11) have been recently identified as potential regulatory targets of endometrial cancer GWAS risk variation through enhancer-promoter chromatin looping studies[23]. Although these data did not support the candidate susceptibility gene HEY2, bioinformatic analysis has previously suggested its targeting by endometrial cancer GWAS risk variation[19].

In terms of the relationships between the candidate susceptibility genes and endometrial cancer, the best-characterized gene is CYP19A1. We have previously linked CYP19A1 to endometrial cancer susceptibility through genetic association with circulating estrogen[19]. CYP19A1 encodes the aromatase enzyme, responsible for a rate-limiting step in the synthesis of estrogen and unopposed estrogen exposure is one of the most well-established risk factors for endometrial cancer[24]. Consistent with its function, increased CYP19A1 expression was associated with increased endometrial cancer risk in our study. The S-MultiXcan-colocalization and MR-JTI analyses provided evidence that adipose-specific expression of CYP19A1 is linked to endometrial cancer risk, concordant with the production of estrogen by

**Table 2 Genes associated with endometrial cancer risk as identified by MR-JTI.**

| Tissue | Locus | Gene | JTI | MR-JTI | | |
|---|---|---|---|---|---|---|
| | | | Z-score | P-value | Beta | 95% CI |
| Subcutaneous adipose | **15q21.2** | **CYP19A1** | **7.03** | **2.00E−12** | **0.35** | **(0.04, 0.57)** |
| | 16p12.1 | NPIPB6 | 4.78 | 1.74E−06 | −0.07 | (−0.23, 0.33) |
| | **17q21.32** | **SNX11** | **4.90** | **9.45E−07** | **0.35** | **(0.04, 0.69)** |
| | 17q21.32 | AC004477.3 | −4.89 | 1.03E−06 | −0.33 | (−0.66, 0.04) |
| Visceral omentum adipose | **15q21.2** | **CYP19A1** | **6.40** | **1.51E−10** | **0.29** | **(0.04, 0.50)** |
| | **17q21.32** | **SNX11** | **4.86** | **1.19E−06** | **0.58** | **(0.38, 0.83)** |
| Ovary | 6q22.31 | HEY2 | −5.82 | 5.89E−09 | 0.13 | (−0.12, 0.36) |
| | 15q15.1 | AC021755.2 | 4.74 | 2.09E−06 | −0.04 | (−0.27, 0.37) |
| | 15q15.1 | AC021755.3 | 4.92 | 8.61E−07 | −0.01 | (−0.31, 0.35) |
| | 15q15.1 | EIF2AK4 | 5.09 | 3.64E−07 | −0.08 | (−0.36, 0.24) |
| | 17q21.32 | AC004477.1 | 4.56 | 5.20E−06 | 0.02 | (−0.34, 0.25) |
| | **17q21.32** | **SNX11** | **4.64** | **3.43E−06** | **0.52** | **(0.25B 0.81)** |
| Uterus | **15q15.1** | **AC021755.3** | **4.78** | **1.77E−06** | **0.32** | **(0.10, 0.54)** |
| Vagina | **3q21.3** | **EEFSEC** | **4.61** | **4.00E−06** | **0.36** | **(0.20, 0.62)** |
| | **15q15.1** | **EIF2AK4** | **5.31** | **1.07E−07** | **0.29** | **(0.09, 0.49)** |
| | 17q21.32 | SNX11 | 4.52 | 6.14E−06 | 0.08 | (−0.17, 0.35) |
| Whole blood | 3q21.3 | EEFSEC | 4.66 | 3.20E−06 | 0.13 | (−0.11, 0.40) |
| | 15q21.2 | CYP19A1 | 6.03 | 1.61E−09 | 0.09 | (−0.08, 0.42) |
| | 17q21.32 | SKAP1 | −4.63 | 3.61E−06 | −0.09 | (−0.37, 0.26) |
| | **17q21.32** | **SNX11** | **4.73** | **2.22E−06** | **0.40** | **(0.20, 0.81)** |

Bonferroni corrected $P$ value thresholds were: $3.7 \times 10^{-6}$ (13,526 genes tested) in JTI$_{subcutaneous\ adipose}$; $3.9 \times 10^{-6}$ (12,949 genes tested) in JTI$_{visceral\ omentum\ adipose}$; $5.8 \times 10^{-6}$ (8615 genes tested) in JTI$_{Ovary}$; $7.1 \times 10^{-6}$ (7049 genes tested) in JTI$_{uterus}$; $7.0 \times 10^{-6}$ (7191 genes tested) in JTI$_{vagina}$ and $4.9 \times 10^{-6}$ (10,140 genes tested) in JTI$_{whole\ blood}$. Bolded findings have evidence of causality in MR-JTI analyses.

aromatase in the adipose of postmenopausal women[25]. Furthermore, drug repurposing analysis linked *CYP19A1* to aromatase inhibitors (letrozole, anastrozole, exemestane and aminoglutethimide) which are used to treat breast cancer. Notably, letrozole has shown efficacy in treating endometrial hyperplasia[26,27], a precursor to endometrial cancer; however, there is less evidence for the use of aromatase inhibitors to treat advanced endometrial cancer[28–30].

S-MultiXcan-colocalization and MR-JTI analyses also suggested tissue-specific associations for expression of *AC021755.3*, *EIF2AK4 HEY2*, and *SKAP1*. The association of *AC021755.3* with endometrial cancer risk was specific to predicted gene expression in the uterus, an organ whose lining consists of the endometrium. *AC021755.3*, a long non-coding RNA with unknown function, is located ~43 kb downstream of *EIF2AK4* and has not been previously related to endometrial cancer. The MR-JTI association of *EIF2AK4* was specific to predicted gene expression in the vagina although it should be noted that the JTI analysis showed nominally significant associations in both uterus and ovary (Supplementary Table 6). The S-MultiXcan-colocalization result for *HEY2* was only significant in the ovary and it is interesting to note that *HEY2* may play a role in the development of ovarian follicles[31,32], which are an important source of sex hormones in premenopausal women. Lastly, the colocalisation of the S-MultiXcan *SKAP1* association in whole blood is consistent with the function of the encoded protein which regulates T-cell receptor signaling[33] and enhances conjugation of T-cells and antigen-presenting cells[34].

PheWAS analysis of the candidate endometrial cancer susceptibility genes revealed associations with traits belonging to seven phenotypic categories, potentially yielding insight into endometrial cancer aetiology. Indeed, five of the categories relate to genetically established endometrial cancer risk factors[2,17–20]. The remaining categories (hematopoiesis and liver function) have not been previously linked with endometrial cancer risk. However, evidence suggests that traits related to red blood cells and platelets, including red blood cell distribution width (found in the

current study), may provide diagnostic markers for endometrial cancer[35]. Furthermore, it was notable to find alkaline phosphatase among the liver traits as intestinal-type alkaline phosphatase levels have recently been predicted to be causally associated with endometrial cancer risk[36] and we also observed a nominally significant genetic corrleation between alkaline phosphatase and endometrial cancer risk. GWAS data for such traits could be used to assess their effects on endometrial cancer risk using similar MR approaches to our studies of body mass index and blood lipids[2,17]. The PheWAS analysis may also provide information about the function of the candidate endometrial cancer susceptibility genes. For example, little is known about *EVI2A* but it was associated with traits such as SHBG and testosterone levels, which have been genetically related to endometrial cancer risk[20], suggesting a role in hormone regulation.

We found four compounds that may counteract gene expression changes associated with endometrial cancer susceptibility. As drug targets supported by genetic evidence of disease association are more likely to receive clinical approval[37,38], there is a strong rationale for future pre-clinical endometrial cancer sensitivity studies of candidate compounds identified in this analysis. One of the compounds, enzastaurin, has already demonstrated anti-cancer effects in endometrial cancer cell lines and thus may have the potential for drug repurposing. Indeed, it has shown clinical benefit in the treatment of serous endometrial cancer in a phase I study of solid tumors[39]. It is also noteworthy that the tubulin inhibitor nocodazole was identified, as the tubulin inhibitor paclitaxel is currently used to treat endometrial cancer patients with advanced disease[40].

We acknowledge some limitations to this study such as small eQTL sample sizes, in particular for uterus, arguably the most relevant tissue for investigating endometrial cancer. However, we have used two approaches that leverage genetic regulation shared across tissues to improve the prediction of gene expression in relevant tissues. Furthermore, we have applied colocalization and MR-JTI analyses to identify candidate causal gene expression associations with endometrial cancer risk, two of which were

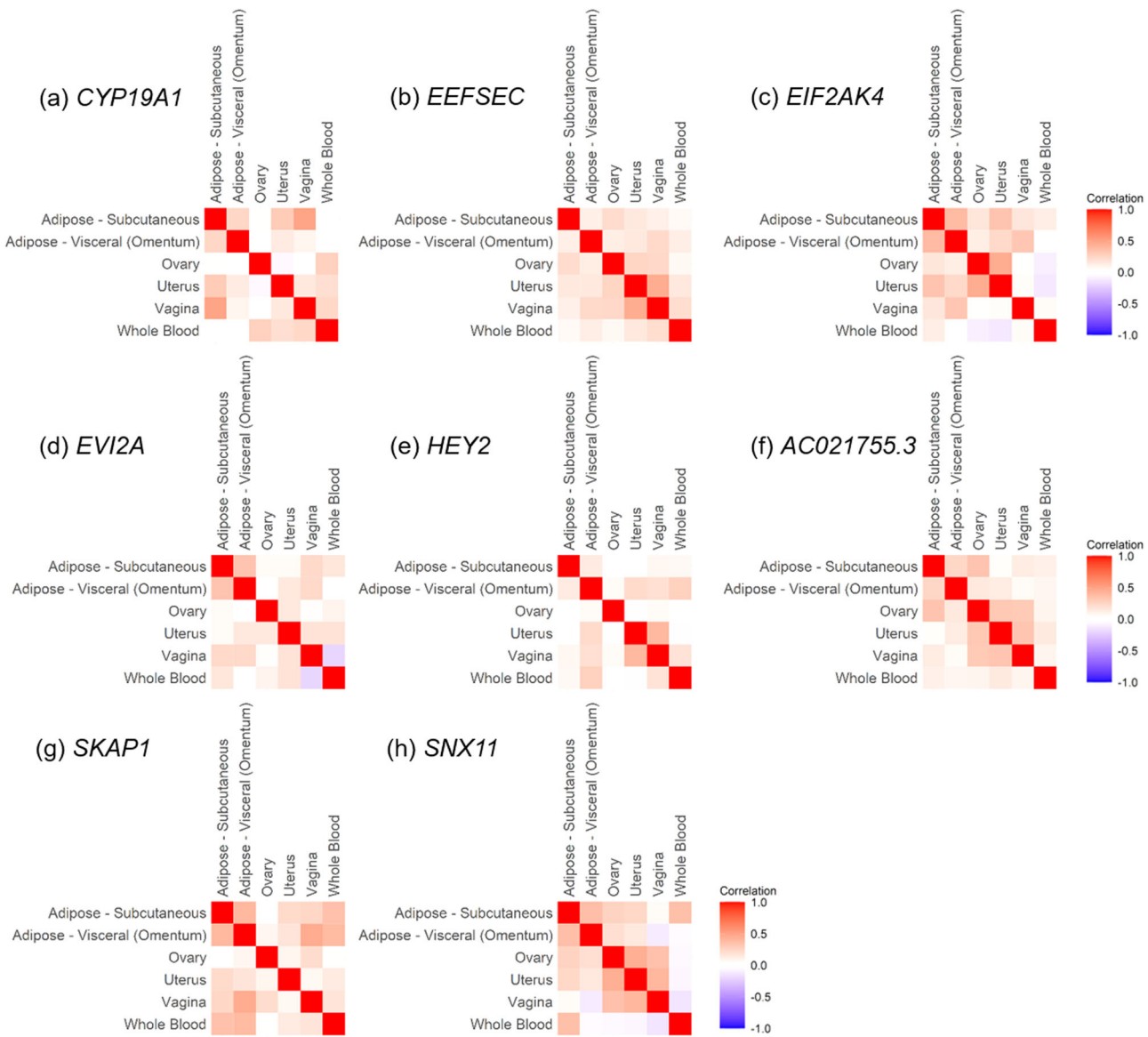

**Fig. 1 Expression correlation of candidate endometrial cancer susceptibility genes between tissues.** Gene expression of *CYP19A1* (**a**), *EEFSEC* (**b**), *EIF2AK4* (**c**), *EVI2A* (**d**), *HEY2* (**e**), *AC021755.3* (**f**), *SKAP1* (**g**) and *SNX11* (**h**) was compared across the six relevant tissues: whole blood, vagina, uterus, ovary, visceral (omentum) adipose and subcutaneous adipose (available in Supplementary Data 1). The intensity of the color shows the magnitude of the expression correlation between tissues, where intense red color indicates a strong positive correlation and intense blue color indicates a strong negative correlation.

**Table 3 Connectivity Map compounds with gene expression profiles opposing endometrial cancer risk TWAS associations.**

| Connectivity score | Compound | Drug class | Clinically tested[*] (highest phase) | Indications |
|---|---|---|---|---|
| −93.60 | Methyl-2,5-dihydroxycinnamate | EGFR inhibitor | No | – |
| −91.07 | Enzastaurin | Protein kinase C inhibitor | Yes (Phase III) | B-cell lymphoma and glioblastoma |
| −90.09 | Prunetin | ABCG2 inhibitor | No | – |
| −90.00 | Nocodazole | Tubulin inhibitor | No | – |

[*]ClinicalTrials.gov database (https://www.clinicaltrials.gov) accessed June 2021.

prioritized by both analyses, and reduce false-positive findings. Differences in the results between the two analyses could be related to the genetic variants used: colocalization analysis was performed using all variants in the gene vicinity whereas MR-JTI

analysis was performed using independent variants ($r^2 < 0.01$) in the vicinity. Thus, the extent to which the colocalization/MR-JTI results represent genuine causal relationships remains unclear. Further functional studies will be necessary to determine the

biological effects of the candidate susceptibility genes identified in this study on experimental phenotypes related to endometrial cancer risk.

In summary, our study has highlighted the value of using a multi-tissue TWAS, with priortization for cauaslity, to identify candidate endometrial cancer susceptibility genes, including biolgolically relevant tissue-specific findings. As well as generating insights into biological mechanisms underlying endometrial cancer risk, and identifying a potential novel risk locus, we have highlighted compounds with potential for endometrial cancer repurposing and provided avenues for future studies of endometrial cancer aetiology.

## Methods

**Endometrial cancer risk GWAS data**. We used summary data from the largest endometrial cancer GWAS meta-analysis to date, comprising 12,906 cases and 108,979 controls of European ancestry[2,41]. All participants provided informed consent and these studies are approved by research ethics committees from QIMR Berghofer Medical Research Institute, University-Clinic Erlangen, Karolinska Institutet, UZ Leuven, The Mayo Clinic, The Hunter New England Health District, The Regional Committees for Medical and Health Research Ethics Norway, and the UK National Research Ethics Service (04/Q0803/148 and 05/MRE05/1). Endometrial cancer subtype analyses were additionally performed using GWAS summary data restricting cases to endometrioid endometrial cancer patients (8758 cases and 46,126 controls)[42]. Detailed descriptions of the quality control procedures and GWAS analysis can be found in the corresponding publication[2]. Briefly, genetic variants with MAF < 1%, genotyping calling rate <98%, genotype missingness above 10%, and deviation from Hardy–Weinberg equilibrium ($P < 10^{-12}$ in cases and $P < 10^{-7}$ in controls) were excluded from the GWAS analysis. After quality control exclusion, 9.5 million variants remained in the GWAS analysis. For TWAS analysis, only variants located outside the major histocompatibility complex region (chr6:26Mb–34 Mb), were used.

**eQTL data**. We used cis-eQTL summary statistics and gene expression data from subcutaneous adipose, visceral omentum adipose, ovary, uterus, vagina and whole blood that are available in the GTEx Project (version 8). Cis-eQTL summary statistics for these tissues and information about respective sample sizes are available from the GTEx website (https://gtexportal.org/home/).

**Multi-tissue TWAS analysis**. S-MultiXcan aggregates single-tissue TWAS results to test the joint effects of gene expression variations from multiple tissues on the trait. To perform S-MultiXcan, we used S-PrediXcan[11] to incorporate endometrial cancer GWAS and eQTL data from each of the six relevant GTEx tissues (version 8) and identify genes associated with endometrial cancer risk. Pre-specified weights based on covariance of genetic variants were used in S-PrediXcan analysis and obtained from PredictDB data repository (http://predictdb.org/). S-PrediXcan data from all tissues were then jointly analysed using multivariate regression in S-MultiXcan[11]. JTI leverages shared genetic regulation across tissues to improve the prediction performance of gene expression in each tissue[16]. For this analysis, we incorporated eQTL data from each of the six relevant GTEx tissues using pre-computed weights based on the covariance of genetic variants[43]. Bonferroni correction was applied to identify statistically significant TWAS genes from S-MultiXcan and JTI. S-PrediXcan, S-MultiXcan and JTI analyses of endometrial cancer GWAS were performed using SPrediXcan.py that was made available on Github (https://github.com/hakyimlab/MetaXcan).

**Colocalization and MR analyses**. As the residual LD amongst genetic variants used to generate gene expression weights in S-MultiXcan and JTI could induce spurious association on nearby non-causal genes, we performed analyses to assess the co-occurrence of endometrial cancer risk and eQTL signals. These subsequent analyses enabled us to tease apart pleiotropic associations (where the same genetic variant is associated with gene expression and disease risk) from linkage (where correlated genetic variants are associated with gene expression and disease risk independently). For S-MultiXcan associations, colocalization analysis was performed using the COLOC package with default parameters in R[15] and the input was identical to the TWAS analysis but restricted to variants within 1 Mb up- and down-stream of the S-MultiXcan identified genes. Full summary statistics of these genes from relevant tissues were retrieved from GTEx portal (version 8 release; https://gtexportal.org/home/). COLOC provided posterior probabilities for the GWAS and eQTL signals to be explained by the same genetic variant (Hypothesis 4, colocalization). We considered a posterior probability for colocalization >0.80 for pleiotropic association as evidence of colocalization within a locus. For JTI associations we performed MR-JTI[16], using the code made available by Gamazon Lab on Github (https://github.com/gamazonlab/MR-JTI). The input was similar to colocalization analysis but further restricted to independent variants (LD < 0.01) that are associated with the expression of JTI identified genes. Full summary statistics for these JTI identified genes were also retrieved from the GTEx portal. LD scores for each independent variant were obtained from pre-computed LD scores

of 1000 Genome European ancestry samples that are made available in BOLT-LMM[44]. Bonferroni adjustment was applied for multiple test correction. Genes that passed S-MultiXcan-colocalization or MR-JTI were prioritized as candidate endometrial cancer susceptibility genes.

**Cross-tissue gene expression correlations**. Gene expression (transcripts per million) data from subcutaneous adipose, visceral omentum adipose, ovary, uterus, vagina, and whole blood were downloaded from the GTEx (v8) project. Pairwise correlations of expression of individual genes across tissues were calculated by the base R function "cor" using pairwise complete observations (i.e., use = "pairwise.complete.obs").

**PheWAS analyses**. We assessed pleiotropic associations of the candidate endometrial cancer susceptibility genes by performing a PheWAS analysis in the CTG-VIEW database[45]. The PheWAS analsyis was performed using summary-data-based MR[46] results from ~1600 phenotypes that are available from this platform. Analyses were performed using eQTL data from relevant or well-powered tissues (i.e., subcutaneous adipose, visceral omentum adipose, ovary, uterus, vagina, and whole blood) and included a heterogeneity test (HEIDI) to identify candidate genes that are affected by the same GWAS risk variant, comparable to our colocalization analysis. Genes passing a Bonferroni correction ($P_{SMR} < 3.1 \times 10^{-5}$ accounting for 1600 phenotypes examined) and no evidence of heterogeneity ($P_{HEIDI} > 0.05$) were defined as having PheWAS associations with candidate susceptibility genes. To explore the shared genetics between endometrial cancer and PheWAS traits, pairwise genetic correlations were performed using LD score regression. Publicly available GWAS summary statistics for endometrial cancer and PheWAS traits were used in this analysis and were derived from O'Mara et al.[2] and the Neale Laboratory (http://www.nealelab.is/uk-biobank), respectively. Bonferroni correction was used to evaluate the statistical significance of the genetic correlation result.

**Computational drug repurposing analysis**. We used the Connectivity Map platform[13] to identify compounds with gene expression profiles opposing the endometrial cancer risk TWAS data. As Connectivity Map analysis requires at least ten upregulated and ten downregulated coding genes as input, we relaxed the threshold in S-MultiXcan and JTI associations to a false discovery rate (FDR)-adjusted P-value < 0.1 to include sufficient unique upregulated and downregulated ($n = 10$ of each) genes for Connectivity Map analysis (Supplementary Tables 6 and 7). Connectivity scores were generated based on a modified Kolmogorov-Smirnov score, which summarises the relationship of the endometrial cancer TWAS gene expression profile with the drug-induced gene expression profile across cancer cell types in the Connectivity Map database. A connectivity score ≤ −90 suggest that the expression of the query genes opposes the drug-induced gene expression profile. We also used the Open Targets platform[14] to determine if candidate endometrial cancer susceptibility genes encode known targets for drugs that have been clinically studied.

**Statistics and reproducibility**. Statistical analyses were primarily performed using Python (https://www.python.org/), R (https://www.r-project.org/), and Plink (https://www.cog-genomics.org/plink/). Bonferroni correction was used to account for multiple testing in multi-tissue TWAS analyses, colocalization and MR analyses, and PheWAS analyses.

**Reporting summary**. Further information on research design is available in the Nature Research Reporting Summary linked to this article.

## Data availability

Summary-level GWAS meta-analysis results for endometrial cancer that support the findings of this study are available at the NHGRI-EBI GWAS Catalog (endometrial cancer: https://www.ebi.ac.uk/gwas/studies/GCST006464; endometrioid endometrial cancer: https://www.ebi.ac.uk/gwas/studies/GCST006465). Cis-eQTL summary statistics are accessible from the GTEx website (https://gtexportal.org/home/) and gene expresion data for candidate susceptibility genes are available in Supplementary Data 1. Pre-specified gene expression weights were obtained from PredictDB data repository (http://predictdb.org/). Other data generated and/or analyzed during this study are included in this article and its supplementary information files or are available on reasonable request.

## Code availability

We used publicly available software (URLs listed below) in this research: S-PrediXcan and S-MultiXcan (https://github.com/hakyimlab/MetaXcan); COLOC (https://cran.r-project.org/web/packages/coloc/index.html); JTI and MR-JTI (https://github.com/gamazonlab/MR-JTI); and Connectivity Map (https://clue.io/cmap). The computer code used in this research is available at https://github.com/pikfang/TWAS.

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

## Acknowledgements

We thank members of the the Endometrial Cancer Association Consortium for contributing their study data, the many women who participated in these studies, and the numerous institutions and their staff who supported recruitment. A full list of consortium members and acknowledgements can be found in the Supplementary Note. We thank Dr Dan Zhou for his help in interpreting MR-JTI results of this study. This work was supported by a National Health and Medical Research Council (NHMRC) Project Grant (APP1109286). P.F.K. is supported by an Australian Government Research Training Program PhD Scholarship and QIMR Berghofer Postgraduate Top-Up Scholarship. T.O.M. and A.B.S. are supported by an NHMRC Investigator Fellowships (APP1173170 & APP177524).

## Author contributions

P.F.K., T.O.M. and D.M.G. designed the study. P.F.K. and X.W. conducted analyses. P.F.K., T.O.M. and D.M.G. interpreted the results. G.C.P. provided resources for phenome-wide analyses. T.D., E.L.G., R.J.S., A.B.S., T.O.M. and the Endometrial Cancer Association Consortium provided endometrial cancer risk GWAS resources. P.F.K., T.O.M. and D.M.G. drafted the manuscript. T.O.M. and D.M.G. supervised the study. A.B.S. and T.O.M. acquired funding. All authors provided critical review of the manuscript.

## Competing interests

The authors declare no competing interests.

**Additional information**

