## [Transparent Peer Review File · Communications Biology]

Reviewers' comments:

Reviewer #1 (Remarks to the Author):

Enclosed is a revision of Kho et al's manuscript:

"Multi-tissue transcriptome-wide association study identifies genetic mechanisms underlying endometrial cancer susceptibility."

In this manuscript, the authors conduct a multi-tissue transcriptome-wide association analysis study of risk of endometrial cancer, building off a recent GWAS for the same phenotype. The authors then ran a PheWAS to understand if TWAS-identified genes showed effects on traits genetically associated with endometrial cancer susceptibility. Lastly, the authors integrated TWAS results with drug repurposing analysis to identify compounds with profiles opposing expression of genes associated with endometrial cancer risk. Overall, the paper is well-motivated, well-written, and clearly communicated. I appreciate that the authors chose an analysis route that uses publicly available data.

I have a few overarching comments and suggestions about the paper: I would like to see a little more motivation, analysis, and discussion about the tissues that lead to the significant TWAS and PheWAS results. It's worth a discussion, as well, if more recent multi-tissue methods may boost power in this study. I've arranged my thoughts into major and minor comments, arranged by section.

Major comments

Abstract

1. The Abstract highlights results that are consistent with previous literature. The author should summarize results that are novel to this study of risk of endometrial cancer.

Results

1. It's my understanding that using multiple tissues tend to select for eQTL architecture that points to gene regulation that is conserved across tissues. A good supplemental figure would be to show the correlations between expression of genes across tissues using the samples that have expression measurements across different tissues. I was wondering if there's a way to show if the authors could add some analysis or discussion into the tissues that point to the susceptibility genes detected.

2. I would like to see some level of analysis about genetic correlations between the multiple phenotypes considered or significantly detected in the PheWAS. That would help contextualize shared genetics between these traits. UKBB summary statistics could be a good place to start.

Methods

1. The rationale for using the Benjamini-Hochberg correction is not valid. The Benjamini-Hochberg correction is valid for positively dependent test statistics (see Benjamini and Yekutieli 2001, The Annals of Statistics). I would adjust the explanation given.

2. Just for clarification: is there a reason why more recent methods like UTMOST (<https://www.nature.com/articles/s41588-019-0345-7?platform=hootsuite>) or MR-JTI (<https://www.nature.com/articles/s41588-020-0706-2>). Specifically, the MRJTI paper points out that MultXcan takes into account multiple tissues only in the association analysis portion of the TWAS approach; JTI, on the other hand, aims to improve prediction, as well. This approach may boost TWAS power in this case, as well, by allowing for more genes to be tested. The weights for GTEx version 8

are also available online for this method. I would suggest the authors run a quick scan using MR-JTI to compare results.

3. In the spirit of open and reproducible science, I would like sample code for analyses provided in a repository.

Minor comments

Introduction

1. I would add a citation to Gusev et al 2016 in Nature Genetics (<https://www.nature.com/articles/ng.3506>) in this line: "As eQTL data are not typically available for GWAS samples, the transcriptome-wide association study (TWAS) approach has been developed to integrate eQTL and GWAS data from independent sample sets..." The Gamazon 2015 paper introduced PrediXcan but the TWAS framework that considered causal mechanisms and incorporated GWAS summary statistics was introduced by this Gusev et al 2016 paper.

Results

1. The shorthand $F_{DR} < 0.05$ should be defined at least once as F_{DR} -adjusted P-value.

Methods

1. Why was F_{DR} -adjusted $P < 0.15$ used as the cutoff for the computational drug repurposing analysis?

Reviewer #2 (Remarks to the Author):

Review: Multi-tissue transcriptome-wide association study identifies genetic mechanisms underlying endometrial cancer susceptibility

Overview:

Kho et al describe their results of a multi-tissue TWAS for endometrial cancer and a subtype of endometrial cancer. Their results are supported by secondary colocalization analysis and PheWAS for identified candidate genes. The manuscript is well written and the results straightforward to follow. I have a few comments.

Major Comments:

1. Can the authors report the tissue enrichment of TWAS signal? The authors support their multi-tissue approach by citing previous works demonstrating sharing of eQTLs, but I'm curious how the distribution of endometrial cancer risk looks across the predicted genes in any given tissue. One crude way to estimate this would be to compute the squared TWAS Z scores and perform a regression against the tissue design matrix, including a mean term. Using R notation, $\text{lm}(z_sq \sim 1 + T)$, where "z_sq" are the squared TWAS z-scores for m tested gene/tissue pairs, and T is the (m x 48) 0/1 design matrix indicating which tissue was used. There will be some correlation between z_sq at nearby genes, so a robust estimate of standard errors can be achieved using bootstrap or something similar.

Minor Comments:

1. The y-axis label in Figure 2 (and Supp Figure 2) should probably be $-\log(p)$ rather than p' .
2. The details regarding quality control, or data prep, of the GWAS are somewhat lacking. Can the authors indicate whether SNPs were filtered based on MAF, HWE, etc?
3. GTEx v8 should have much larger sample sizes and improve power, but this is a minor point given current work climate and totally understand if the authors prefer not to include this suggestion at this

stage.

Response to editor and reviewers

Editor's comments

- (1) Examine tissue-specific enrichment of TWAS results and generate correlations of gene expression across tissue types, as suggested by both referees. On a related note, we would ask that you examine genetic correlations between significant PheWAS traits, as suggested by Referee #1.**

To examine tissue-specific enrichment of TWAS results, we have performed MR-JTI analysis (**Table 2**) as suggested by Reviewer #1, in addition to restricting the original S-MultiXcan analysis to six relevant tissues (subcutaneous adipose, visceral omentum adipose, ovary, uterus, vagina and whole blood) and using the subsequent colocalisation analysis to also provide evidence of tissue specificity (**Table 1**).

Correlations of candidate endometrial cancer susceptibility gene expression across the six relevant tissues have been added to the manuscript (**Figure 1**):

Genetic correlations between significant PheWAS traits and endometrial cancer are now presented in **Supplementary Table 4** and pair-wise genetic correlations with PheWAS traits are shown in **Supplementary Figure 1**:

(2) If feasible, we highly encourage you to update your analysis using GTEx v8 data, as suggested by Referee #2. At a minimum, we would recommend that you perform the comparison with MR-JTI (using GTEx v8 data) suggested by Referee #1.

We have updated our S-MultiXcan analysis using GTEx v8 data (**Table 1**). We also performed the MR-JTI analysis using GTEx v8 data (**Table 2**) and found two candidate endometrial cancer

susceptibility genes that were also among the five genes prioritized by colocalisation of the S-MultiXcan findings (**lines 136-138**):

Two candidate endometrial cancer susceptibility genes (CYP19A1 and EEFSEC) were prioritized using both S-MultiXcan-colocalization and MR-JTI approaches.

(3) Carefully proofread and further discuss the novelty, context, and methods of the study, as noted by both referees.

We have further discussed the methods (**see highlighted sections in Methods**) and novelty of the study (**see highlighted sections in Abstract and comment #4 below**), in particular the intriguing tissue-specific findings produced by the new analyses, and attempted to put these findings into a biological context (**lines 217-229**):

S-MultiXcan-colocalization and MR-JTI analyses also suggested tissue-specific associations for expression of AC021755.3, EIF2AK4 HEY2, and SKAP1. The association of AC021755.3 with endometrial cancer risk was specific to predicted gene expression in the uterus, an organ whose lining consists of endometrium. AC021755.3, a long non-coding RNA with unknown function, is located ~43 kb downstream of EIF2AK4 and has not been previously related to endometrial cancer. The MR-JTI association of EIF2AK4 was specific to predicted gene expression in the vagina although it should be noted that the JTI analysis showed nominally significant associations in both uterus and ovary (Supplementary Table 6). The S-MultiXcan-colocalization result for HEY2 was only significant in the ovary and it is interesting to note that HEY2 may play a role in the development of ovarian follicles (Terauchi, Shigeta, Iguchi, & Sato, 2016; Trombly, Woodruff, & Mayo, 2009), which are an important source of sex hormones in premenopausal women. Lastly, the colocalisation of the S-MultiXcan SKAP1 association in whole blood is consistent with the function of the encoded protein which regulates T-cell receptor signaling (Wu, Fu, & Shen, 2002) and enhances conjugation of T-cells and antigen-presenting cells (Wang et al., 2003).

Reviewer #1 comments

(4) The Abstract highlights results that are consistent with previous literature. The author should summarize results that are novel to this study of risk of endometrial cancer.

We have added a summary of the novel results to abstract, particularly those from the revised analysis (**lines 56-64**):

... we have used the largest endometrial cancer GWAS and gene expression from six relevant tissues to maximize statistical power, prioritizing for causality eight candidate endometrial cancer susceptibility genes: AC021755.3, EEFSEC, EIF2AK4, CYP19A1, HEY2, SNX11, EVI2A and SKAP1. Notably, AC021755.3 and EEFSEC have not previously been identified as candidate endometrial cancer susceptibility genes and EEFSEC is located at a potentially novel endometrial cancer risk locus. We also show evidence of biologically relevant tissue specificity for associations

of endometrial cancer susceptibility with predicted expression of CYP19A1 (adipose), HEY2 (ovary) and SKAP1 (whole blood).

- (5) It's my understanding that using multiple tissues tend to select for eQTL architecture that points to gene regulation that is conserved across tissues. A good supplemental figures would be show the correlations between expression of genes across tissues using the samples that have expression measurements across different tissues. I was wondering if there's a way to show I was wondering if the authors could add some analysis or discussion into the tissues that point to the susceptibility genes detected.**

We have added correlations of the expression of candidate endometrial cancer susceptibility genes identified in this study in **Figure 1**. We have included this as a main figure as we thought that these findings were intriguing (i.e. gene expression is generally positively correlated across the solid tissues but less so in whole blood) (**lines 150-153**):

Gene expression of the candidate susceptibility genes was generally positively correlated across the solid tissues studied (Figure 1). However, gene expression in whole blood was occasionally inversely, or less well correlated, with expression levels in other tissues (e.g. EVI2A, EIF2AK4 and SNX11).

We have also revised the tissues analysed to a set of six which we think are most relevant or related to endometrial cancer development and the subsequent colocalization of the S-MultiXcan results in these tissues provides some evidence of specificity (**Table 1**). In addition, we have performed MR-JTI TWAS analysis (**Table 2**) which generates tissue-focused results, allowing us to further identify potentially tissue-specific results. These findings are discussed in **lines 217-229** (see reply to comment #3 above for text).

- (6) I would like to see some level of analysis about genetic correlations between the multiple phenotypes considered or significantly detected in the PheWAS. That would help contextualize shared genetics between these traits. UKBB summary statistics could be a good place to start.**

We have shown genetic correlation between PheWAS traits in a heat map (**Supplementary Figure 1**, displayed in reply to comment #1 above)

- (7) The rationale for using the Benjamini-Hochberg correction is not valid. The Benjamini-Hochberg correction is valid for positively dependent test statistics (see Benjamini and Yekutieli 2001, The Annals of Statistics). I would adjust the explanation given.**

We have analysed data using the more stringent Bonferroni correction to control for false positive rates in our findings (**lines 309-312**):

Bonferroni correction was applied to identify statistically significant TWAS genes from S-MultiXcan and JTI. S-PrediXcan, S-MultiXcan and JTI analyses of endometrial cancer GWAS

were performed using *SPrediXcan.py* that was made available on Github (<https://github.com/hakyimlab/MetaXcan>).

However, we used $FDR < 0.1$ to provide enough genes for the computational drug repurposing analysis (lines 358-361):

...we relaxed the threshold in S-MultiXcan and JTI associations to a false discovery rate (FDR)-adjusted P-value < 0.1 to include sufficient unique upregulated and downregulated (n=10 of each) genes for Connectivity Map analysis (Supplementary Tables 6 and 7).

(8) Just for clarification: is there a reason why more recent methods like UTMOST (<https://www.nature.com/articles/s41588-019-0345-7?platform=hootsuite>) or MR-JTI (<https://www.nature.com/articles/s41588-020-0706-2>). Specifically, the MRJTI paper points out that MultiXcan takes into account multiple tissues only in the association analysis portion of the TWAS approach; JTI, on the other hand, aims to improve prediction, as well. This approach may boost TWAS power in this case, as well, by allowing for more genes to be tested. The weights for GTEx version 8 are also available online for this method. I would suggest the authors run a quick scan using MR-JTI to compare results.

We have updated the S-MultiXcan analysis using GTEx v8 data (Table 1), and we have also added MR-JTI results to the manuscript (Table 2). Two of the five genes prioritized by the colocalization of the S-MultiXcan results were also found by the MR-JTI analysis (see reply to comment #2).

(9) In the spirit of open and reproducible science, I would like sample code for analyses provided in a repository.

Computer code used in this study is available at <https://github.com/pikfang/TWAS> (see Code Availability, lines 376-381):

We used publicly available software (URLs listed below) in this research: S-PrediXcan and S-MultiXcan <https://github.com/hakyimlab/MetaXcan>; COLOC <https://cran.r-project.org/web/packages/coloc/index.html>; JTI and MR-JTI <https://github.com/gamazonlab/MR-JTI>; Connectivity Map <https://clue.io/cmap>; Open Targets Platform <https://platform.opentargets.org/>. The computer code used in this research is available at <https://github.com/pikfang/TWAS>.

(10) I would add a citation to Gusev et al 2016 in Nature Genetics (<https://www.nature.com/articles/ng.3506>) in this line: “As eQTL data are not typically available for GWAS samples, the transcriptome-wide association study (TWAS) approach has been developed to integrate eQTL and GWAS data from independent sample sets...” The Gamazon 2015 paper introduced PrediXcan but the TWAS framework that considered causal mechanisms and incorporated GWAS summary statistics was introduced by this Gusev et al 2016 paper.

We have updated the citation in **line 86**.

(11) The shorthand $FDR < 0.05$ should be defined at least once as FDR-adjusted P-value.

We have revised this definition in **line 359**.

(12) Why was FDR-adjusted $P < 0.15$ used as the cutoff for the computational drug repurposing analysis?

The Connectivity Map analysis requires at least 10 upregulated and 10 downregulated protein coding genes as input, whereas we have identified only seven protein coding genes from S-MultiXcan and JTI analyses using a Bonferroni correction. Thus, we have relaxed the threshold in S-MultiXcan and JTI associations to $FDR < 0.10$ to perform Connectivity Map analysis. We have added this explanation to the Methods (**lines 359-361**, see reply to comment #7 for text).

Reviewer #2 comments

(13) Can the authors report the tissue enrichment of TWAS signal? The authors support their multi-tissue approach by citing previous works demonstrating sharing of eQTLs, but I'm curious how the distribution of endometrial cancer risk looks across the predicted genes in any given tissue. One crude way to estimate this would be to compute the squared TWAS Z scores and perform a regression against the tissue design matrix, including a mean term. Using R notation, $lm(z_sq \sim 1 + T)$, where "z_sq" are the squared TWAS z-scores for m tested gene/tissue pairs, and T is the (m x 48) 0/1 design matrix indicating which tissue was used. There will be some correlation between z_sq at nearby genes, so a robust estimate of standard errors can be achieved using bootstrap or something similar.

We have revised the tissues analysed to a set of six that we think are most relevant to endometrial cancer development and the subsequent colocalization of the S-MultiXcan results in these tissues provides some evidence of specificity as reported (**lines 123-124**):

The colocalization data also suggested tissue-specific associations for CYP19A1 (subcutaneous adipose), HEY2 (ovary) and SKAP1 (whole blood) (Table 1).

We have also performed the MR-JTI analysis to identify TWAS signals which are enriched in specific tissues (**lines 133-135**):

These analyses indicated some tissue specificity in the predicted gene expression associations: CYP19A1 in adipose (subcutaneous and visceral omentum), AC021755.3 in uterus, and EEFSEC and EIF2AK4 in vagina.

For expression of the eight candidate endometrial cancer susceptibility genes identified between the two approaches we have provided in **Figure 1** (displayed in reply to comment #1) an analysis of their correlation between the six tissues studied.

(14) The y-axis label in Figure 2 (and Supp Figure 2) should probably be ‘-log(p)’ rather than ‘p’.

Figure 2 and Supplementary Figure 2 have been removed from the manuscript, as the information is already presented in **Table 1** and **Supplementary Table 1**.

(15) The details regarding quality control, or data prep, of the GWAS are somewhat lacking. Can the authors indicate whether SNPs were filtered based on MAF, HWE, etc?

We have added further details about quality control of GWAS in Methods (**lines 285-289**):

Briefly, genetic variants with MAF < 1%, genotyping calling rate < 98%, genotype missingness above 10% and deviation from Hardy-Weinberg equilibrium ($P < 10^{-12}$ in cases and $P < 10^{-7}$ in controls) were excluded from the GWAS analysis. After quality control exclusion, 9.5 million variants remained in the GWAS analysis.

(16) GTEx v8 should have much larger sample sizes and improve power, but this is a minor point given current work climate and totally understand if the authors prefer not to include this suggestion at this stage.

We have updated the analyses with GTEx v8 data as described in response to comment #2.

REVIEWERS' COMMENTS:

Reviewer #1 (Remarks to the Author):

I have no further comments for this paper. I commend the authors for including substantial comparison of expression (whole and GReX) across tissues.

Reviewer #2 (Remarks to the Author):

The authors have addressed my previous comments and I have no new comments this time.